# A Framework For Differentiable Discovery Of Graph Algorithms

## Abstract

Recently there is a surge of interests in using graph neural networks (GNNs) to learn algorithms. However, these works focus more on imitating existing algorithms, and are limited in two important aspects: the search space for algorithms is too small and the learned GNN models are not interpretable. To address these issues, we propose a novel framework which enlarge the search space using cheap global information from tree decomposition of the graphs, and can explain the structures of the graph leading to the decision of learned algorithms. We apply our framework to three NP-complete problems on graphs and show that the framework is able to discover effective and explainable algorithms.

## 1 Introduction

Many graph problems such as maximum cut and minimum vertex cover are NP-hard. The classical algorithm design paradigm often requires significant efforts from domain experts to understand and exploit problem structures, in order to come up with effective procedures. However, for more complex problems and in the presence of a family of problem instances, it is becoming increasingly challenging for human to identify the problem structures and tailor algorithms. Thus there is a surge of interests in recent years to use learning and differentiable search to discover graph algorithms.

In this context, GNNs have been widely used for representing and learning graph algorithms (Dai et al., 2018; Li et al., 2018). However, directly using a GNN model to define the algorithm search space may not be enough for discovering an algorithm better than existing greedy ones. Hella et al. (2015); Sato et al. (2019) have theoretically discussed the limitations of GNNs for expressing more powerful algorithms, by bridging the connection between GNNs and distributed local algorithms. Sato et al. (2019) derived the approximation ratios of the algorithms that can be learned by GNNs, which are much worse than those of some simple algorithms (Johnson, 1974; Chlebík & Chlebíková, 2008). Intuitively, GNNs can only capture local graph patterns, but miss out the global information, which fundamentally restricts their expressiveness power.

To enhance the capacity of GNNs and allow for a larger search space, we incorporate global information about the graph as additional features, and augment them with other node/edge features. The idea of incorporating additional features to improve the expressiveness of GNNs has been deployed in serval existing models, by adding either unique node identifiers (Donnat et al., 2018; Seo et al., 2019; Zhang et al., 2020), the information of port numbering (Sato et al., 2019), or randomness (Sato et al., 2020). However, these features are added mainly to break the local symmetry of similar graph patterns, but do not add much information about the global graph properties.

Another important aspect which have largely been ignored in previous work is explaining the learned algorithm encoded in GNN. Many previous works focus on the ability of GNNs to imitate existing graph algorithms, without showing new algorithms are being learned. One exception is that Khalil et al. (2017) experimentally showed that GNN has discovered a new algorithm for minimum vertex cover problem where the node selection policy balances between node degree and the connectivity of the graph. However, this phenomenon was just mentioned in passing, and a systematic explanation of the graph patterns leading to the algorithm decision is missing. Therefore, there is an urgent need to develop explainable graph models to understand the learned algorithm.

In this paper, we propose a new framework for differentiable graph algorithm discovery (DAD), focusing on two important aspects of the discovery process: designing a larger search space, and an effective explainer model. More specifically, we design a search space for graph algorithms by

Figure 1: DAD has 3 components: (1) augment problem instance graph with cheap global information; (2) learn graph neural networks with augmented graphs; (3) explain the learned GNN with a graph explainer model.

*augmenting GNNs with cheap global graph features*. Our proposed global features are specially designed for learning graph algorithms. It reveals to be a simple yet effective way of enhancing the capacity of GNNs. After the GNN is sufficiently trained, we employ a novel *graph explainer model* to figure out the structural property of the graph leading to the algorithm decision. The graph explainer model allows us to transform the algorithm represented in continuous embedding space back to the interpretable discrete space, so that it can assist human experts in understanding the discovery.

We demonstrate our framework on three NP-hard graph algorithms: minimum vertex cover (MVC), maximum cut (Max-Cut), and minimum dominating set (MDS). Our DAD framework is able to discover graph algorithms that achieve better approximation ratios. Besides, we apply our explainer model to the learned algorithms and demonstrate some interesting and explainable patterns.

Except for methodology, we also contribute in generating a new dataset. To conduct experiments on these three problems, we need to generate the solutions for the purpose of evaluation or supervised learning. We run `Gurobi` for around 12,000,000 core hours to generate a reasonably large set of (graph problem, solution) pairs, with varying graph sizes. It could be a very useful benchmark dataset for graph algorithm learning, which will help with future researches in this area.

## 2 OVERVIEW

Our framework will innovate on three important aspects of the graph algorithm discovery process. That is, how to design a larger search space, how to learn the GNN algorithms, and how to effectively explain the GNN after it is learned. An overview of our framework is illustrated in Figure 1.

**Search space design.** To allow GNNs to represent a more powerful space of algorithms, we make use of cheap global information which can be obtained in subquadratic time in the number of nodes and edges of the graph. In particular, we will find spanning trees of the original graph, and use the solutions on these trees as augmented node/edge features. In this way, the capacity of GNNs can be enhanced, and a larger space will be searched over to discovery new and effective algorithms.

**Learning method.** We can use either supervised or unsupervised approaches to train the GNNs. In supervised setting, the goal is to learn a GNN algorithm that can imitate the results of expensive solvers but run much faster. In unsupervised setting, a larger unlabeled dataset can be used for training, and it is favorable for large graph problems where supervisions are hard to obtain.

**Explainer model.** We design a novel graph explainer model based on an information theoretic formulation, where we train a subgraph pattern selection model such that the selected subgraph patterns are more influential to the decisions of the original GNN model in an information theoretic sense. Furthermore, the explainer model is trained by differentiating through a blackbox combinatorial solver. Since our selection model produces discrete and interpretable local graph structures as explanations, it provides human intuition on what the kind of new algorithm is learned.

## 3 DIFFERENTIABLE GRAPH ALGORITHM DISCOVERY (DAD) FRAMEWORK

### 3.1 BACKGROUND

We focus on three NP-hard problems in this paper: MVC, Max-Cut, and MDS. Let $G = (V, E, w)$ be a weighted graph, where $V$ denotes the set of nodes, $E$ the set of edges, and $w : E \mapsto \mathbb{R}$ the edge weight function. Furthermore, let $\boldsymbol{A}$ and $\boldsymbol{L}_w$ be the adjacent and weighted Laplacian matrices of $G$, and $\boldsymbol{B} \in \{0,1\}^{|E| \times |V|}$ be the incidence matrix, where $\boldsymbol{B}_{ei} = 1$ and $\boldsymbol{B}_{ej} = 1$ if $e = (i,j) \in E$ and

otherwise 0. Then these problems can be formulated as integer programming (IP) problems:

$$\min_{\boldsymbol{y}} f(\boldsymbol{y}; G) \quad \text{subject to} \quad \boldsymbol{g}(\boldsymbol{y}; G) \leq \boldsymbol{0} \text{ and } \boldsymbol{y} \in \{0, 1\}^{|V|}, \tag{1}$$

where the objective and the set of constraints $\boldsymbol{g} = [g_1, \cdots, g_l]^\top$ are listed respectively in the table:

The optimal solutions $\boldsymbol{y}^* \in \{0, 1\}^{|V|}$ to the above problems can be viewed as binary labels on each node. Therefore, learning these solutions can be modeled as binary node classification tasks. Furthermore, we will denote $Y = [Y_1, \cdots, Y_{|V|}]^\top \in \{0, 1\}^{|V|}$ as the random variable representing the labels on the nodes, and $\boldsymbol{y} = [y_1, \cdots, y_{|V|}]^\top$ as the values of the random variable $Y$.

|  | $\lvert f(\boldsymbol{y}; G) \rvert$ | $\boldsymbol{g}(\boldsymbol{y}; G)$ |
|---|---|---|
| MVC | $\mathbf{1}^\top \boldsymbol{y}$ | $\mathbf{1} - B\boldsymbol{y}$ |
| Max-Cut | $\boldsymbol{y}^\top \boldsymbol{L}_w \boldsymbol{y}$ | $\mathbf{0}$ |
| MDS | $\mathbf{1}^\top \boldsymbol{y}$ | $\mathbf{1} - \boldsymbol{y} - A\boldsymbol{y}$ |

### 3.2 Cheap Solutions As Global Features

What global features could improve the expressive power of GNNs for solving graph problems? The following example motivates our design of global features.

**A motivating example.** While the best known algorithm for solving a general linear system takes time $\mathcal{O}(n^{2.373})$ (Williams, 2012), Kelner et al. (2013) proposed a simple algorithm for solving the system $\boldsymbol{L}_w \boldsymbol{x} = \boldsymbol{b}$ where $\boldsymbol{L}_w$ is the Laplacian matrix of a graph $G$, in nearly-linear time. The algorithm can be summarized by 2 major steps: (i) Find a low-stretch spanning tree $T \subseteq G$, and solve the problem on the spanning tree $\boldsymbol{L}_T \boldsymbol{x}_0 = \boldsymbol{b}$ to obtain an initial solution $\boldsymbol{x}_0$; (ii) Refine the initialized solution by iteratively operating on local cycles in the original graph $G$. The authors also formally proved the convergence and run-time guarantees for the algorithm.

Inspired by the above successful algorithm with theoretical guarantees, we generalize the idea to solve a broader class of graph problems. In many graph problems, global optimal solutions can be found efficiently by algorithms such as dynamic programming if the graph is a tree. Therefore, we propose to use **(i) solutions on spanning trees** $T \subseteq G$ and **(ii) approximate solutions of greedy algorithms on** $G$ as cheap global features to augment the original graphs. More specifically,

(i) Given a graph $G$, we first find its spanning tree $T \subseteq G$. There are multiple ways of constructing the trees, and we can obtain $n$ spanning trees $T(1), \cdots, T(n)$ for each graph, using the same set of $n$ algorithms. On each tree $T(k)$, we can apply dynamic programming (DP) based algorithms to quickly find an optimal solution on the tree, denoted by a binary vector $\boldsymbol{y}^{T(k)} \in \{0, 1\}^{|V|}$. The tree solutions, together with the trees themselves, can be used as global features for the graph $G$.

$$T(1), \cdots, T(n) \xmapsto[\text{dynamic programming}]{\text{global optimal on spanning trees}} \{(T(1), \boldsymbol{y}^{T(1)}), \cdots, (T(n), \boldsymbol{y}^{T(n)})\} \tag{2}$$

(ii) We can also operate on the original graph $G$ to quickly obtain an approximate solution by using some greedy algorithms. With a set of $m$ greedy algorithms denoted by $\text{Gd}(1), \cdots, \text{Gd}(m)$, we can obtain a set of approximate solutions.

$$\text{Gd}(1), \cdots, \text{Gd}(m) \xmapsto[\text{different greedy algorithms}]{\text{approximate solution}} \{\boldsymbol{y}^{\text{Gd}(1)}, \cdots, \boldsymbol{y}^{\text{Gd}(m)}\} \tag{3}$$

The tree solutions and greedy solutions will be concatenated and used as node features. Besides, each spanning tree can be represented by a binary matrix $\boldsymbol{T}(k) \in \{0, 1\}^{|V| \times |V|}$, where the $(i, j)$-th entry indicates whether the edge $(i, j)$ is in the tree. These spanning trees will also be concatenated and used as edge features. Then the overall features can be denoted by

**node features:** $\boldsymbol{X} := [\boldsymbol{y}^{T(1)}, \cdots, \boldsymbol{y}^{T(n)}, \boldsymbol{y}^{\text{Gd}(1)}, \cdots, \boldsymbol{y}^{\text{Gd}(m)}] \in \{0, 1\}^{|V| \times (n+m)}, \tag{4}$

**edge features:** $\boldsymbol{Z} := [\boldsymbol{T}(1); \cdots; \boldsymbol{T}(n)] \in \{0, 1\}^{|V| \times |V| \times n}. \tag{5}$

In the rest of this paper, we denote $\boldsymbol{X}_i$ as the $i$-th row vector of $\boldsymbol{X}$ which corresponds to the features of node $i$, and $\boldsymbol{Z}_{ij}$ as the vector $\boldsymbol{Z}[i, j, :]$ which corresponds to the features of the edge $(i, j)$.

We note that these tree and greedy solution features can typically be computed in time linear in the number of nodes and edges, which will not change substantially the computational complexity of the GNN algorithms learned later. Furthermore, in our later experiments, we also show that the proposed global features can be generally applied to different graph problems and performs better than other trivially augmented features.

### 3.3 LEARNING DISTRIBUTED LOCAL GRAPH ALGORITHM

Given a graph $G = (V, E, w)$ and its associated global features $\boldsymbol{X}$ and $\boldsymbol{Z}$, a GNN computes the node embeddings by iteratively aggregating the messages from neighbors:

$$\boldsymbol{h}_i^{(t+1)} \leftarrow \boldsymbol{F}_\theta \left( \boldsymbol{h}_i^{(t)}, \boldsymbol{X}_i, \{\boldsymbol{h}_j^{(t)}, \boldsymbol{X}_j, \boldsymbol{Z}_{ij}\}_{j \in \mathcal{N}(i)} \right), \tag{6}$$

where $\boldsymbol{h}_i^{(t)}$ is the $d$-dimensional feature embedding for node $i \in V$ at $t$-th iteration, $\theta \in \Theta$ are the parameters in the GNN, $\boldsymbol{F}_\theta$ is a nonlinear propagation function, and $\mathcal{N}(i)$ denotes the neighbors of node $i$. Note that for a weighted graph we could also use the edge weights as edge features. In the rest of this paper, we overload the notation $\boldsymbol{Z}$ and assume it is the concatenation of the global feature $\boldsymbol{Z}$ and the weights of the graph if provided.

After $T$ iterations, each node embedding $\boldsymbol{h}_i^{(T)}$ will contain information about its $T$-hop neighborhood. An output layer is applied to compute the probability distribution of $Y_i$:

$$Y_i \sim \text{Bernoulli}(p_i), \quad p_i = \texttt{OutputLayer}(\boldsymbol{h}_i^{(T)}), \quad \forall i \in V. \tag{7}$$

Given the predictive distribution $\{p_i\}_{i \in V}$, we will retrieve the binary solution $\{y_i \in \{0,1\}\}_{i \in V}$ by a method of conditional expectation described in (Raghavan, 1988; Karalias & Loukas, 2020).

To learn the parameters $\theta$ in the GNN, we consider both the supervised and unsupervised setting:

**Supervised learning with ILP solvers.** In the supervised setting, for each graph $G$, a solution $\boldsymbol{y}^*$ is given by running an expensive solver, and the goal is to learn a GNN-based distributed local algorithm that can imitate the results of the expensive solver but runs much faster. However, one challenge is that there could exist symmetry in the solution $\boldsymbol{y}^*$. For example, given a feasible solution $\boldsymbol{y}^*$ to a Max-Cut problem, the vector $1 - \boldsymbol{y}^*$ is an equivalent solution that gives the same cost and the same cut-set, so there could be multiple modes in the labels which will cause the 'mode-averaging' problem (Hinton et al., 1995). To overcome this problem, instead of predicting only one probability $p_i$ as in Eq. 7, we generate $K$ many probability distributions, and employ the hindsight loss for each node prediction (Guzman-Rivera et al., 2012; Li et al., 2018). That is

$$\text{Hindsight loss: } \min_{k \in \{1, \cdots, K\}} \ell(y_i^*, p_{i,k}) \quad \text{where} \quad p_{i,k} = \texttt{OutputLayer}_k(\boldsymbol{h}_i^{(T)}), \tag{8}$$

$K$ could be the number of modes, and $\ell$ is the binary cross entropy loss.

**Unsupervised learning with continuous relaxation.** Generating optimal solutions for large graphs can be very expensive. Therefore, we also consider the unsupervised setting which allows the use of a large unlabeled training dataset. We will construct the unsupervised training loss based on the optimization objective $f$ and constraints $\boldsymbol{g} = [g_1, \cdots, g_l]$ of the graph problems described in Sec 3.1. The design of the loss mainly follows (Karalias & Loukas, 2020) and more details and principles can be found there. To summarize, let $\boldsymbol{p} = \{p_i\}_{i \in V}$ be the predictive probability given by the GNN, our unsupervised loss is defined as the following probabilistic penalty loss:

$$\mathcal{L}_U(\boldsymbol{p}, G) := \mathbb{E}[f(Y; G)] + \beta \cdot \sum_{i=1}^l \mathbb{P}[g_i(Y; G) \leq 0], \quad \text{where} \quad Y \sim \text{Bernoulli}(\boldsymbol{p}), \tag{9}$$

and $\beta$ is a penalty coefficient. Note that it is a relaxed version of the loss proposed by Karalias & Loukas (2020), who considers the probability of the satisfaction of all constraints jointly. Here we treat the $l$ constraints separately for efficient computations. See Appendix B for more details.

**Approximation ratio:** As GNN is a type of distributed local algorithm, it is easy to see that it admits the identify function in the function space. Given that we can use the greedy algorithmic features, it is easy to see that our design space admits an algorithm that enjoys the same approximation ratio as the greedy algorithm, like $\mathcal{O}(\log n)$ for MDS, or 2 for MVC and Max-Cut. This naturally achieves or improves the ratio obtained by Sato et al. (2019).

### 3.4 EXPLAIN THE LEARNED GNN

We simply denoted the learned GNN as $F_\theta$. To understand what algorithm is discovered in $F_\theta$, we propose a novel method to explain its predictions $\{p_i\}_{i \in V}$. We want to find that, for each node $i$, which subset of nodes $V_S^i \subseteq V$ and edges $E_S^i \subseteq E$ are most influential to the prediction $p_i$. Besides, the global features $X$ and $Z$ are induced by the solutions of $n$ spanning trees and $m$ greedy algorithms, which we denote as $\mathcal{S} := \{T(1), \cdots, T(n), \texttt{Gd}(1), \cdots, \texttt{Gd}(m)\}$. We are also interested in finding which trees or algorithms provide more useful features for the prediction.

Since we focus on node-level predictions, our explainer is defined as a mapping that finds node-dependent subsets

$$\text{Explainer:} \quad \mathcal{E} : (i, V, E, \mathcal{S}) \mapsto \left(V_S^i, E_S^i, \mathcal{S}_S^i\right), \tag{10}$$

where $V_S^i$ and $E_S^i$ are selected subsets of $k_V$ nodes and $k_E$ edges respectively, and $\mathcal{S}_S^i \subseteq \mathcal{S}$ is a subset of $k_{\mathcal{S}}$ trees/algorithms. More precisely, we parameterize $\mathcal{E}$ as a second GNN along with a top-$K$ operation, where the GNN is applied to score the set of structural elements in $(V, E, \mathcal{S})$, and the top-$K$ operation selects the $(k_V, k_E, k_{\mathcal{S}})$ highest scoring elements. Details are provided below.

**Parameterization of $\mathcal{E}$.** It is notable the prediction $p_i$ is computed by $F_\theta$ based on the $T$-hop subgraph of the node $i$ and is irrelevant to nodes of distance larger than $T$. Therefore, when working on node $i$, the explainer only needs to select the influential nodes/edges from its $T$-hop subgraph, denoted by $G(i, T) := (V(i, T), E(i, T), w)$. Therefore, we define explainer GNN as

$$\text{Explainer GNN:} \quad F_\phi^{\mathcal{E}} : (V(i, T), E(i, T), X(i, T), Z(i, T)) \mapsto \{\boldsymbol{h}_j^{\mathcal{E}}\}_{j \in V(i,T)}, \tag{11}$$

where $X(i, T) := [X_j]_{j \in V(i,T)}$ and $Z(i, T) := [Z_{jk}]_{(j,k) \in E(i,T)}$ are features induced by the subgraph. Then multi-layer perceptrons (MLP) are applied to define the scoring functions:

(I) Node scores: $c_j = \texttt{MLP}_\phi^V(\boldsymbol{h}_j^{\mathcal{E}})$ for each node $j \in V(i, T)$.
(II) Edge scores: $c_{j,k} = \texttt{MLP}_\phi^E([\boldsymbol{h}_j^{\mathcal{E}}, \boldsymbol{h}_k^{\mathcal{E}}, E_{j,k}])$ for each edge $(j, k) \in E(i, T)$.
(III) Feature scores: $\boldsymbol{c} = \texttt{MLP}_\phi^{\mathcal{S}}\left(\texttt{Pooling}\left(\{\boldsymbol{h}_j^{\mathcal{E}}\}_{j \in V(i,T)}\right)\right)$, where the output $\boldsymbol{c}$ is a $(n + m)$-dimensional vector that gives the score for features induced by the $n$ spanning trees and the $m$ greedy algorithms.

With the scores, the subsets $(V_S^i, E_S^i, \mathcal{S}_S^i)$ can be selected as choosing the elements with scores in top-$K$, for $K = k_V, k_E$, and $k_{\mathcal{S}}$, respectively. In summary, the explainer model can be written as the composition of three operations

$$\text{Parameterized Explainer:} \quad \mathcal{E}_\phi = \text{top-}K \circ \texttt{MLP}_\phi \circ F_\phi^{\mathcal{E}}. \tag{12}$$

**Variational lower bound of mutual information.** To learn the explainer $\mathcal{E}_\phi$, we employ the learning to explain (L2X) framework in (Chen et al., 2018). In L2X, $\mathcal{E}_\phi$ is learned by maximizing the mutual information between the predicted label $Y_i$ and the selected subsets $(V_S^i, E_S^i, \mathcal{S}_S^i)$. However, due to intractability, a variational distribution parameterized by a neural network $Q_\psi$ will be learned jointly with the explainer $\mathcal{E}_\phi$ to maximize the variational lower bound of the mutual information

$$\max_{\phi, \psi} \; \mathbb{E}\left[\log Q_\psi\left(Y_i \mid V_S^i, E_S^i, \mathcal{S}_S^i\right)\right], \; \text{s.t. } (V_S^i, E_S^i, \mathcal{S}_S^i) = \mathcal{E}_\phi(i, V, E, X, Z), \tag{13}$$

where the expectation is taken over the predictive probability $p_i$ given by the learned GNN, over all node $i$ in graph $G$ and all graph $G$ in the training set $\mathcal{G}_{train}$.

**Parameterization of $Q$.** Again we will parametrize the variational distribution $Q_\psi$ by a third GNN $F_\psi^Q : (V_S^i, E_S^i, X_S^i, Z_S^i) \mapsto \{\boldsymbol{h}_j^Q\}_{j \in V(i,T)}$ followed by pooling and MLP:

$$\text{Variational Distribution:} \quad Q_\psi = \texttt{MLP}_\psi\left(\texttt{Pool}\left(F_\psi^Q(i, V_S^i, E_S^i, X_S^i, Z_S^i)\right)\right), \tag{14}$$

where the features $(X_S^i, Z_S^i) = H(V_S^i, E_S^i, \mathcal{S}_S^i, X, Z)$ are induced by the selected subsets via a differentiable mapping $H$.

**Differentiable top-$K$.** The additional technical challenges for optimizing the variational objective in Eq. 13 is that the top-$K$ operation in the parameterized explainer (Eq. 12) is not differentiable. More specifically, a top-$K$ operation over the scores $c_1, \cdots, c_M$ for some $M$ is equivalent to solving the following integer linear programming (ILP) problem:

$$\arg\max_{\boldsymbol{x} \in \{0,1\}^M} \sum_{j=1}^M c_j x_j \quad \text{s.t. } \sum_{j=1}^M x_j = K. \tag{15}$$

It is easy to see that the optimal solution $x_j^* = 1$ only if $c_j$ is among the top-$K$ scores. Given this equivalence, we can employ the blackbox differentiation techniques introduced by Vlastelica et al. (2019) to compute the gradient of the ILP solver, and optimize the parameters $\phi$ in the explainer.

**Discussion.** Our explainer model is very different from GNNExplainer (Ying et al., 2019), which may be of independent interests in term of explainer model for GNNs. For instance, GNNExplainer optimizes the selection mask (i.e., the explainer) for an individual graph, while we parameterize the

explainer as another GNN which can be applied to explain the predictions on a class of different graphs. Besides, the variational $Q$ in GNNExplainer is defined to be the original GNN $F_\theta$ which is fixed, while we optimize $Q$ over a parametric family, so that our method is more principled and optimal. Furthermore, we formulate the top-$K$ selection as a solution to an ILP and apply a blackbox differentiation technique to optimize the explainer model, which has not been attempted in the space of explainer models before. We provide some empirical justifications in Appendix G. Discussion with other explainer models can be found in Appendix A.

## 4 RELATED WORK

**Expressiveness of GNNs for representing algorithms.** GNNs have been used to learn graph algorithms. Some recent works have discussed the representation power of GNNs for representing graph algorithms. Xu et al. (2018) showed that GNNs are at most as powerful as the Weisfeiler-Lehman (WL) test in distinguishing graph structures. Maron et al. (2019; 2018) proposed $k$-GNN which can represent the results of $k$-WL test. Barceló et al. (2019) showed that GNNs are too weak to represent logical Boolean classifiers - FOC$_2$, and proposed to add a global read-out function break this limitation. Focusing on combinatorial problems including MDS and MVC, Sato et al. (2019) derived the approximation ratios of algorithms that GNNs can learn, and theoretically showed that adding the information of port numbering and weak-coloring can improve those approximation ratios.

**Learning based graph algorithms.** Empirically, several works have demonstrated the ability of GNNs for imitating existing graph algorithms. For instance, GNN is used to learn the convergent states of graph algorithms (Dai et al., 2018), imitate DP algorithms (Xu et al., 2019), and imitate individual steps and intermediate outputs of classical graph algorithms (Veličković et al., 2019). To learn a new and more effective graph algorithm, Khalil et al. (2017) combined reinforcement learning with GNNs to learn greedy policies that can solve graph problems such as MVC and Max-cut. Karalias & Loukas (2020) proposed an unsupervised framework for learning graph algorithms and demonstrated the results on maximum clique and local partitioning problems.

## 5 EXPERIMENT

In Section 5.1, we first introduce problems we are going to tackle, and the baselines we are going to compare with. Then in Section 5.2 and Section 5.3, we show the empirical approximation ratio of DAD on several synthetic and real-world datasets, and compare against several known polynomial algorithms, as well as other relevant graph neural network variants. Finally in Section 5.4, we leverage our proposed explainer to understand the learned algorithm.

### 5.1 EXPERIMENT SETUP

We leave the details of greedy algorithms and dynamic programming on trees in Appendix C.
We first compare against some of the cheap algorithms that are also served as features. As we use the continuous relaxation in Eq. 9, we also compare with the linear/semi-definite programming (LP/SDP) with randomized rounding for these problems:

- MVC: `LogN-Approx`(node-degree greedy), `2Approx-Greedy`(edge selection with degree greedy), `2Approx`(edge selection) and LP relaxation are included.
- MDS: `Greedy`(sequential coverage based greedy selection) and LP relaxation are included.
- Max-CUT: `2Approx-Greedy`(greedy local adjustment), SDP and `MaxSpanning`(solution directly obtained on max spanning tree, as any node assignment is a valid cut) are included.

We here include several recent works that extends GNNs with different forms of global information:

- `Onehot` (Karalias & Loukas, 2020; Seo et al., 2019): adds one-hot encoding to a random node.
- `RandFeat` (Sato et al., 2020): uses random node features to break the tie.
- `CPNGNNs` (Sato et al., 2019): leverages port numbering in GNN's message passing, together with weak 2-coloring node features. As CPNGNNs only works for graphs with bounded degree, we use the position encoding (Vaswani et al., 2017) for port number representation instead.
- `Raw`: we simply use constant (or edge weight, when applicable) for node/edge features, which serves as the baseline for all the global information augmented GNNs.

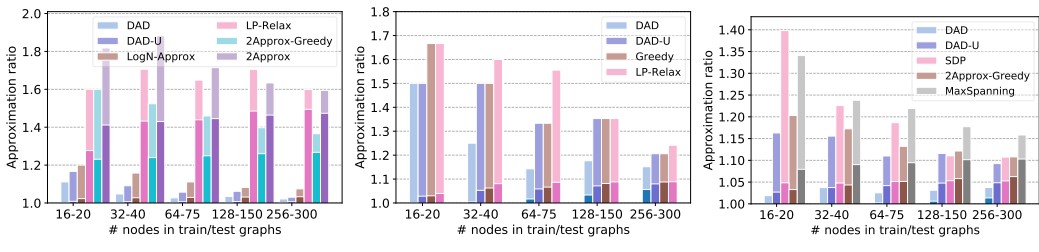

MVC-Barabasi Albert    MDS-Barabasi Albert    Maxcut-Barabasi Albert

Figure 2: Test approximation ratio for different algorithms. Foreground color (darker) shows the average test ratio, while background color (lighter) shows the maximum test ratio.

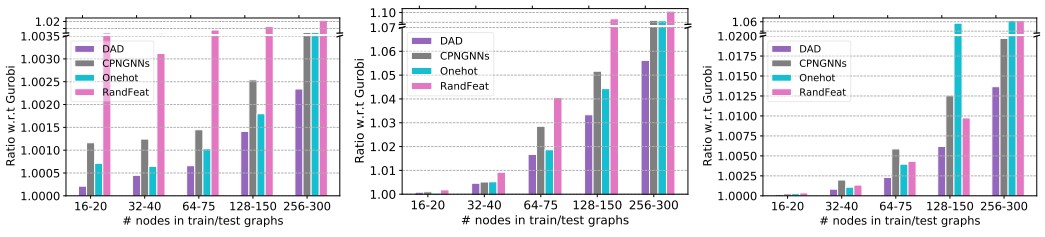

MVC-Barabasi Albert    MDS-Barabasi Albert    Maxcut-Barabasi Albert

Figure 3: Test approximation ratio for GNNs with other types of global information.

## 5.2 QUANTITATIVE RESULTS ON RANDOM GRAPHS

**Data preparation** We use two random graph models, namely Erdos Renyi (ER) with edge probability $0.15$, and Barabasi Albert (BA) with edge attachment 4 to generate random graphs for training and test. We vary the graph sizes in $\{16\text{-}20, 32\text{-}40, 64\text{-}75, 128\text{-}150, 256\text{-}300, 512\text{-}600, 1024\text{-}1100\}$. In each $\{\text{problem, graph type, graph size}\}$ configuration, we generation $\{10k, 1k, 1k\}$ graph instances for training, validation and test, respectively. We run Gurobi on each instance to get the solution for evaluation or supervised training. See Appendix D for more details on setup and Gurobi.

**Main results:** Figure 2 shows the comparison between classical algorithms. We can see DAD leverages but goes beyond the cheap algorithms, with both supervised and unsupervised (DAD-U) learning strategies. The gain is significant regarding both the average and maximum test ratio. While the maximum ratio is similar in MDS, DAD still gains advantages in average case, which is expected as we do empirical risk minimization during training. Figure 3 compares against other GNNs with global information, where our design consistently outperforms alternatives. We also tried to pad the baseline GNN features with either random or constant values to match the number of parameters in GNN. This doesn't help the baselines much, which verifies that the gain of DAD is not simply coming from the input size. See Appendix F.3.1 for more details.

**Extrapolation:** We train the models on graphs up to 300 nodes, and evaluate them up to 1100 nodes to see their extrapolation ability. We show the details in Appendix F.2, where the extrapolation results are still consistently better than the best classical baseline algorithm.

**Ablation study:** Here we try to understand the contributions of tree solution versus greedy solution. We present such study in Figure 12 in appendix on the three tasks. Our main takeaways are: 1) both of the two feature families can boost GNN in most cases, comparing to not using any features. 2) having multiple trees can also be helpful for MVC and MDS, as the spanning tree is not unique. 3) using both types of the features is better than either of them. This validates our design principle.

**Time-Solution trade-off:** As exhaustive algorithms can already achieve optimal solution in finite time, we want to study the trade-off between time and the approximation ratio obtained by different algorithms. We show the results in figure 4 on BA graphs with minimum 1024 nodes. For each single graph, we record down the time spent for solving and the approximation ratio, and obtain a dot in the scatter plot. As Gurobi solver is an iterative process, we plot the data points every time it improves the solution on the graph, and truncate. We can see DAD can achieve similar or better solution quality than what can be achieved by Gurobi, with 2 magnitudes less time.

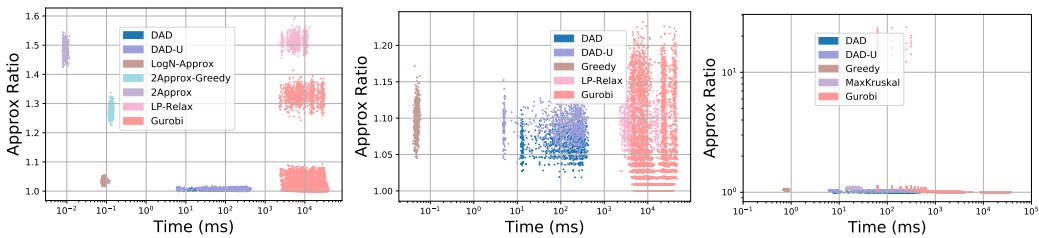

MVC-Barabasi Albert          MDS-Barabasi Albert          Maxcut-Barabasi Albert

Figure 4: Time and approximation ratio trade-off plot for different methods.

Table 1: Real world MVC and MDS test results.

| Data | MVC | | MDS | |
|---|---|---|---|---|
| | 2nd-best baseline | DAD | 2nd-best baseline | DAD |
| Twitter | $1.68e^{-2}$ | $\mathbf{1.93e^{-3}}$ | $6.43e^{-2}$ | $\mathbf{8.42e^{-3}}$ |
| Github | $1.64e^{-3}$ | $\mathbf{1.09e^{-5}}$ | $3.89e^{-4}$ | $\mathbf{4.14e^{-5}}$ |
| Memetracker | $1.81e^{-2}$ | $\mathbf{1.55e^{-5}}$ | $1.30e^{-4}$ | $\mathbf{7.67e^{-5}}$ |

Table 2: Real world MAXCUT.

| | Optsicom-B | Optsicom-G |
|---|---|---|
| 2Approx | $7.65e^{-2}$ | $7.98e^{-2}$ |
| MaxSpanning | $7.63e^{-2}$ | $7.09e^{-2}$ |
| SDP | $8.36e^{-1}$ | $8.52e^{-1}$ |
| DAD | $\mathbf{4.23e^{-3}}$ | $\mathbf{1.16e^{-5}}$ |

## 5.3 RESULTS ON REAL WORLD GRAPHS

**Data preparation** We first collect several social network datasets `Twitter` and `Github` from Morris et al. (2020). We also collect `Memetracker`, `Optsicom` and follow Khalil et al. (2017) to generate training/evaluation graphs. We run MVC and MDS tasks on the social networks, and run MAXCUT on the physics(`Optsicom`) graphs. See Appendix E for more information about the dataset statistics and real-world experiment results.

**Results:** We show the relative error compared to the optimal solution obtained by `Gurobi` in Table 1 and Table 2. We report DAD's performance with supervised learning setting. It still achieves the best performance on both the social networks and physics graphs, where the relative error can be several magnitudes smaller than the second best classical algorithms in some cases.

## 5.4 EXPLAINING LEARNED ALGORITHMS.

In this section, we use our proposed explainer to explain the learned algorithm on BA graphs. By zooming in into the important discrete substructures of the graph, we are able to find some interesting algorithmic patterns that aligns with our intuition.

**Explanation with edge selection:** Here we limit the number of nodes to be 5, and edges to be 10 when explaining each target node of a graph, and parameterize $\mathcal{E}_\phi$ and $Q_\psi$ using GNN with the same number of layers as the target GNN. We found in figure 5 that some nodes depict the greedy behavior, where the budget for edge explanations is mostly allocated to adjacent edges. Note that although we use greedy algorithm solution as features, the GNN doesn't know about the underlying physical meaning of them. This shows that our explainer is able to re-discover the greedy algorithm.

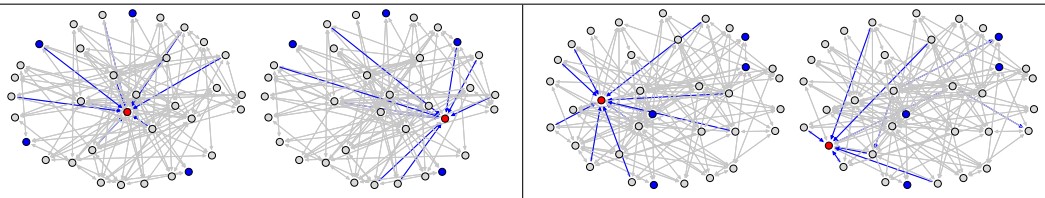

Figure 5: Greedy-like algorithm behavior on some nodes in MVC(left) and MDS(right) tasks. Selected nodes and edges are colored blue, where the target node for explanation is colored red.

**Explanation with node selection:** When running the explainer on a single graph with different target nodes, we found that there are several nodes appear frequently in the selected subset. We call these nodes as *anchor nodes*. We try to understand the correlation between anchor nodes and the node predictive probabilities in figure 6. We first run explainer on every node in a graph, and count how many times each node has appeared in the explanation of other nodes. The node color indicates such frequency, where the dark red has highest frequency. We can clearly see the existence of such anchors. Then we show the predicted node probability as the boundary color of each node, where the dark blue has probability close to 1.0. We can see no matter how many nodes with dark blue boundaries, the anchor nodes tend to distribute evenly among both high probability and low

Figure 6: Anchor nodes v.s. node selection probability. From left to right: MVC, MDS, Max-Cut.

probability nodes. Our hypothesis is that GNN leverage the similarities between target node and the diverse anchor nodes to make a proper decision, where the anchors serve like landmarks in kernel methods. Such *anchor node* behavior is also consistent to some recent distance based designs (You et al., 2019; Li et al., 2020) of GNNs.

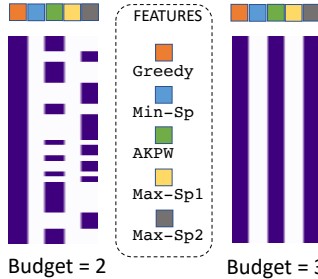

Figure 7: Algorithm selection

**Explanation with feature selection:** We further look at the contribution of different cheap global features to the GNN prediction. We take the Max-Cut problem, where we use 5 global features, including `Greedy` and {Min, AKPW, Max, Max}-Spanning trees. We include `MaxSpanning` twice as the cut solutions are symmetric and come in pairs. We can see from figure 7 when budget is set to 3, all the explanations select { `Greedy`, `AKPW`, `MaxSpanning` }, despite that `MaxSpanning` solution has higher quality than `AKPW`. This shows that the explainer learns that two `MaxSpaning` are duplicated in identifying global information. When the budget is limited to 2, our explainer still consistently identifies the best performing algorithm, *i.e.*, `Greedy`. From this we can learn two principles in instantiating our framework: 1) better algorithm solution generally introduces better global information; 2) having diverse algorithmic solutions is better than multiple similar ones.

## 6 CONCLUSION

In this paper, we propose a novel framework, called DAD, for differentiable discovery of graph algorithms. We have demonstrated that our framework can search over a large space of graph algorithms using GNNs and global features, and can interpret the learned algorithm by a novel explainer model. It can potentially serve as a powerful tool to improve the existing graph-based algorithms, as well as providing explanations for other downstream tasks.

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

# Appendix

## A  OTHER RELATED WORKS

**Explainer models.** Existing work on explainer models approach the problems from three directions. The first line of work use gradients of the outputs with respect to inputs to identify the salient features in the inputs (Simonyan et al., 2013; Springenberg et al., 2014); The second approximates the model with simple interpretable models, such as locally additive models (Bach et al., 2015; Kindermans et al., 2016; Ribeiro et al., 2016; Lundberg & Lee, 2017); the third defines input pattern selection operators, such that the outputs of the model based on the selected input patterns has large mutual informaton with the original model outputs (Chen et al., 2018; Ying et al., 2019). The difference of our explainer model compared to GNNExplainer (Ying et al., 2019) is discussed in Sec 3.4. The explainer we used also shares the similar merits as a concurrent work published by Luo et al. (2020). The main difference lies in the optimization technique, where the stacked Gumbel-softmax is used in Luo et al. (2020), while we use a differentiable top-k operator that is more stable and easier to train in general.

## B  PROBABILISITC RELAXATION OF CONSTRAINTS

In main paper we mentioned that we are solving the following probabilistic relaxation of the integer programming:

$$\mathcal{L}_U(\boldsymbol{p}, G) := \mathbb{E}[f(Y; G)] + \beta \cdot \sum_{i=1}^{l} \mathbb{P}[g_i(Y; G) \leq 0], \quad \text{where} \quad Y \sim \text{Bernoulli}(\boldsymbol{p}),$$

This can be more general for many constrained optimization problems. For example:

**MVC:**  MVC constraint requires that each edge $e = (i, j)$ has $p_i + p_j \geq 1$. In the probabilistic sense, the probability of violating this constraint is $(1 - p_i) * (1 - p_j)$. Thus the expected penalty would be:

$$\beta \sum_{e=(x,y)\in G} (1 - p_x) * (1 - p_y) \tag{16}$$

**MDS:**  MDS constraints are placed on nodes. The ILP constraint says $p_i + \sum_{j \in \mathcal{N}(i)} p_j \geq 1$. Again, we write it in the probability sense, with the expected penalty be:

$$\beta \sum_{i \in V} (1 - p_i) * \prod_{j \in \mathcal{N}(i)} (1 - p_j) \tag{17}$$

In our experiment, we use above expected penalty to handle the hard constraints in unsupervised learning setting.

## C  DETAILS OF CHEAP ALGORITHMIC FEATURES

The cheap algorithm configurations are described in Table 3. For unweighted graphs the spanning

Table 3: Cheap algorithmic configurations for DAD on different problems.

|  | Tree solutions | Greedy Algorithms |
|---|---|---|
| MVC | DP on random spanning tree | Node/Edge based greedy |
| MDS | DP on random spanning tree | Node based greedy |
| Max-Cut | Two-coloring on {max,min,low-stretch} spanning tree | Greedy local adjustment |

trees are usually not unique, so we can use multiple tree solutions as global features for DAD. We use AKPW (Alon et al., 1995) for computing the low-stretch spanning tree for Max-Cut problems.

## C.1 Obtaining tree solutions

Note that although these combinatorial optimization on trees are easier than NP-hard, but they are still nontrivial. Here we cover the recursion of dynamic programming for these problems:

**MVC:**   Firstly we pick a random node as root, and traverse the tree in DFS order. Suppose we use $f[n, 0]$ to denote the cover size of the subtree rooted at node $n$, when the node $n$ is not chosen, and $f[n, 1]$ to denote the cover size when node $n$ is chosen, then we have:

$$
\begin{aligned}
f[n, 0] &= \sum_{c \in \text{child(n)}} f[c, 1] \\
f[n, 1] &= 1 + \sum_{c \in \text{child(n)}} \min\{f[c, 0], f[c, 1]\}
\end{aligned} \tag{18}
$$

And the final cover size is obtained by $\min\{f[\text{root}, 0], f[\text{root}, 1]\}$.

**MDS:**   Firstly we pick a random node as root, and traverse the tree in DFS order. Suppose we use $f[n, 0]$ to denote the dominant set size of the subtree rooted at node $n$, when the node $n$ is chosen; use $f[n, 1]$ to denote the situation when $n$ is not picked, but it has been dominated by other nodes; use $f[n, 2]$ to denote that we don't care about node $n$'s situation, but its subtree should satisfy the constraint. Then we have:

$$
\begin{aligned}
f[n, 0] &= 1 + \sum_{c \in \text{child(n)}} \min\{f[c, 0], f[c, 1], f[c, 2]\} \\
f[n, 2] &= \sum_{c \in \text{child(n)}} \min\{f[c, 0], f[c, 1]\} \\
f[n, 1] &= \begin{cases} \infty, \text{child(n)} = \emptyset \\ f[n, 2] + \min_{c \in \text{child(n)}}\left[f[c, 0] - \min\{f[c, 0], f[c, 1]\}\right], \text{ otherwise} \end{cases}
\end{aligned} \tag{19}
$$

and finally the dominant set size equals to $\min\{f[\text{root}, 0], f[\text{root}, 1]\}$.

**Max-Cut:**   Max-Cut on tree is easy when all the edge weights are non-negative, as every tree is a bipartite graph. We can perform two-coloring on this tree to obtain the solution. By flipping the colors afterwards, we can get another equivalent solution. Although this solution is simple, it still has nontrivial effect on obtaining the global information, as the 2-coloring solution on tree is the weak 2-coloring on the original graph. It is known that for some problems, such information can help boost the approximation ratio of distributed local algorithms (Sato et al., 2019).

## D   Experimental details

We run all the experiments on a cluster with 6149 nodes. In each node, there are 64 cores. To obtain the solution for evaluation or supervised training, for each graph per task, we run `Gurobi` on one node with the time limit as 1 hour. As we have three tasks, two graph models, seven graph sizes, and $12k$ graph instances in each category, the computational budget is 32 million core hours. In reality, we used around 12 million core hours because only the larger graph tasks can run up to 1 hour.

Compared to that of the exact solution, the core hour consumption of the relaxed LP is neglectable. However, the SDP for Max-CUT is computationally intensive. We used `cvxpy` to realize it. For the larger graphs, it can take up to 40 mins to resolve one graph with 64 cores. It takes around 0.2 million core hours to obtain the SDP solutions.

We tuned the hyperparameters of GNN models for each graph category, such as the number of layers and the number of spanning trees, using grid search. We run each job up to 24 hours on a 64-core node. It takes around 5 million core hours to run the jobs on the synthetic datasets and 0.3 million core hours for the real datasets.

In total, it takes us around 17.5 million core hours to run all the jobs.

|                | Collab  | Twitter | Github | Facebook | Memetracker | Optsicom |
|----------------|---------|---------|--------|----------|-------------|----------|
| Average # nodes | 74.49  | 131.76  | 83.83  | 81.66    | 315.36      | 125.0    |
| Average # edges | 2457.22 | 1709.32 | 117.32 | 80.58   | 335.68      | 375.0    |
| # train        | 3,000   | 583     | 7,635  | 597      | 10,000      | 10,000   |
| # valid        | 1,000   | 194     | 2,545  | 199      | 1,000       | 1,000    |
| # test         | 1,000   | 196     | 2,545  | 199      | 1,000       | 1,000    |

Table 4: Real world data statistics.

Table 5: Real world MVC and MDS.

| Data | MVC | | | | | MDS | | |
|------|-----|-----|-----|-----|-----|-----|-----|-----|
|      | LogN-Approx | 2Approx-Greedy | 2Approx | LP-Relax | DAD | Greedy | LP-Relax | DAD |
| Collab | $1.22e^{-3}$ | $8.17e^{-2}$ | $1.06e^{-1}$ | $1.21e^{-1}$ | $\mathbf{1.28e^{-4}}$ | 0 | 0 | 0 |
| Facebook | 0 | 0 | 0 | 0 | 0 | $5.32e^{-3}$ | $\mathbf{0}$ | $\mathbf{0}$ |
| Twitter | $1.68e^{-2}$ | $1.62e^{-1}$ | $2.88e^{-1}$ | $3.84e^{-1}$ | $\mathbf{1.93e^{-3}}$ | $8.26e^{-2}$ | $6.43e^{-2}$ | $\mathbf{8.42e^{-3}}$ |
| Github | $5.61e^{-3}$ | $5.52e^{-1}$ | $8.23e^{-1}$ | $1.64e^{-3}$ | $\mathbf{1.09e^{-5}}$ | $6.24e^{-3}$ | $3.89e^{-4}$ | $\mathbf{4.14e^{-5}}$ |
| Memetracker | $1.81e^{-2}$ | $4.50e^{-1}$ | $7.42e^{-1}$ | $6.09e^{-1}$ | $\mathbf{1.55e^{-5}}$ | $2.38e^{-2}$ | $1.30e^{-4}$ | $\mathbf{7.67e^{-5}}$ |

## E  MORE REAL WORLD DATA INFORMATION

We collect a set of social network dataset Collab, Twitter, Github and Facebook from Morris et al. (2020), following Karalias & Loukas (2020). Each dataset comes with a set of graphs, and we split them into training/validation/test with ratio 6:2:2. Additionally, we collect Memetracker graph from SNAP (Leskovec & Krevl, 2014) and Optsicom used in Khalil et al. (2017). As these only have a few graphs, we use the same strategy from Khalil et al. (2017) to generate graphs. Specifically, for Memetracker, we use widely-adopted Independent Cascade model and sample a diffusion cascade from the full graph with constant set to 7, and consider the largest connected component in the graph as a single graph instance; for Optsicom we keep the graph structures, but perturb the edges weights using Bernoulli distribution (denoted as Optsicom-B) or Gaussian distribution with zero mean and 0.1 standard deviation (denoted as Optsicom-G).

See Table 4 for the final statistics about the datasets.

For the completeness, we additionally include MVC and MDS results on Collab and Facebook datasets in Table 5. These datasets are used in Karalias & Loukas (2020) but not very suitable for evaluating MVC or MDS tasks, as it is easy to get exact solution with simple algorithms on thest two datasets.

## F  MORE EXPERIMENTAL RESULTS

### F.1  MORE IN-DISTRIBUTION GENERALIZATION RESULTS

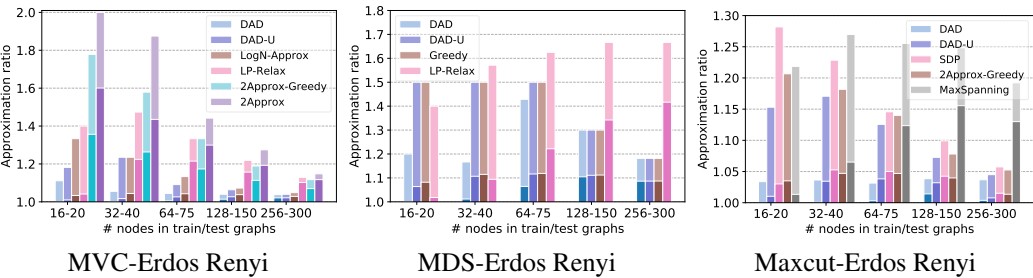

MVC-Erdos Renyi          MDS-Erdos Renyi          Maxcut-Erdos Renyi

Figure 8: Test performance comparison with other classical algorithms on Erdos Renyi graphs.

In main paper we covered the comparison between DAD and other classical algorithms, as well as other types of global GNNs on Barabasi Albert graphs. Here we present the results on Erdos Renyi

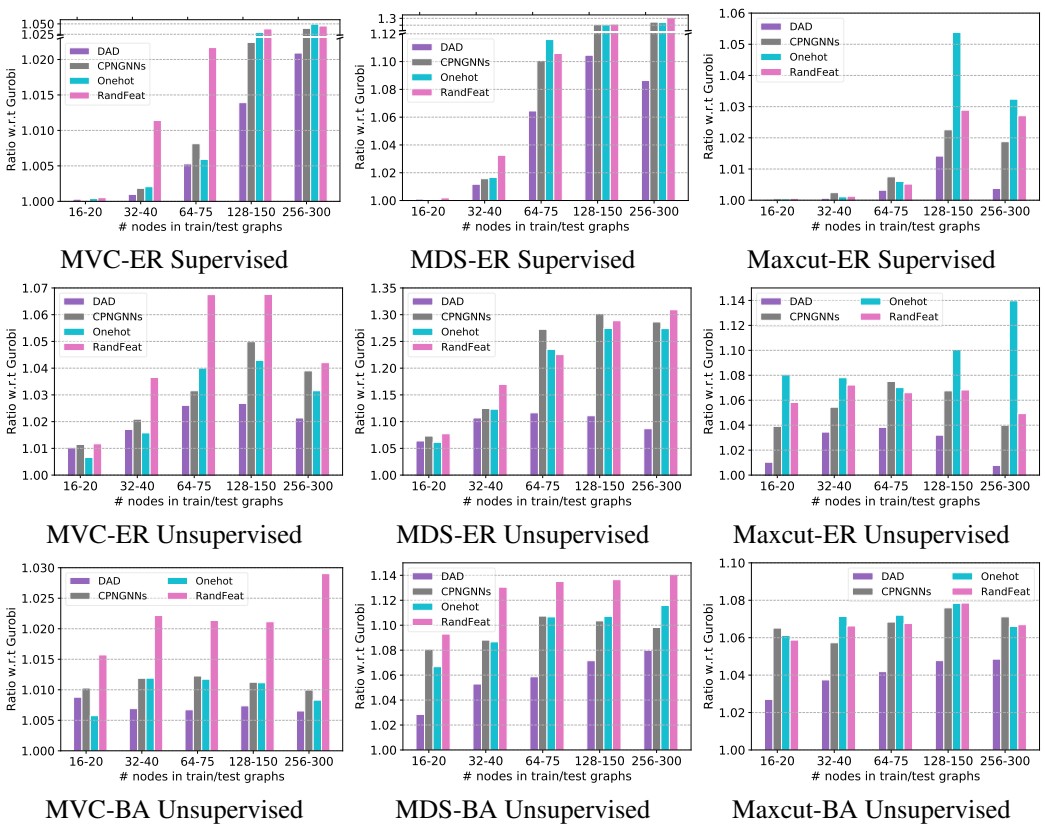

Figure 9: Test approximation ratio for GNNs with other types of global information.

graphs, as well as the results with unsupervised learning in figure 8 and figure 9. We can see the proposed DAD consistently beats all other alternatives in all these settings.

## F.2    MORE EXTRAPOLATION RESULTS

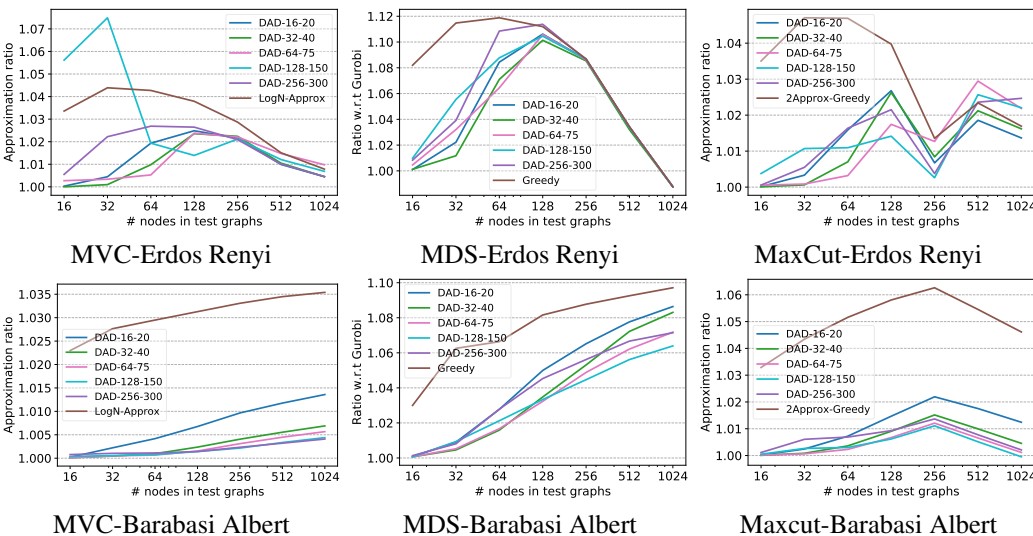

Figure 10: Extrapolation results with supervised learning, compared with the best polynomial baseline (greedy).

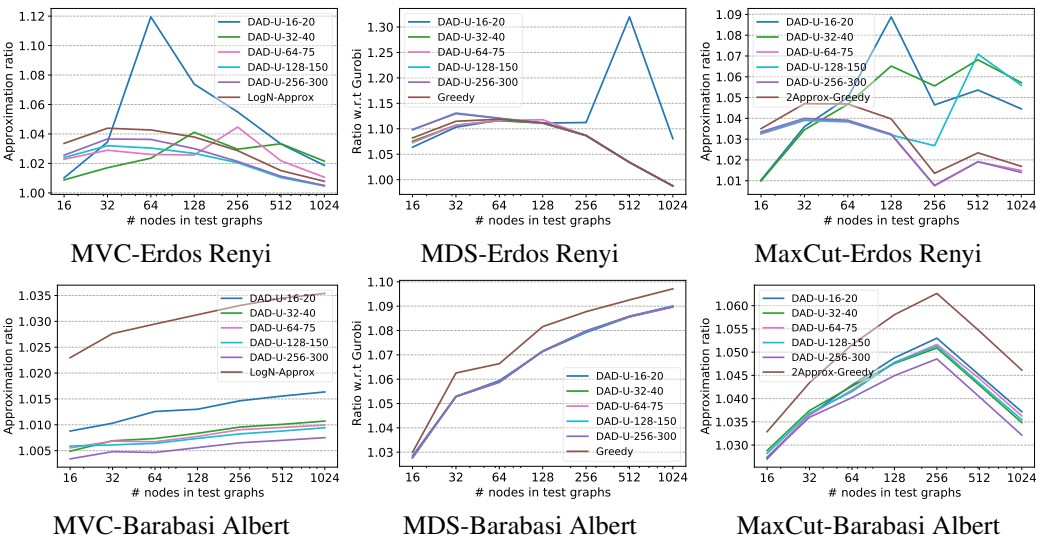

Figure 11: Extrapolation results with unsupervised learning, compared with the best polynomial baseline (greedy).

Here we show the extrapolation results on Erdos Renyi and Barabasi Albert graphs, where DAD is trained with supervion on graphs with up to 300 nodes. We can see from figure 10, DAD still extrapolates well to large graphs.

The unsupervised learning extrapolates in a similar way, as seen in figure 11.

One interesting finding is that, it achieves approximation ratio that is less than 1 on MDS large graphs. This indicates that the `Gurobi` is not capable of generating high quality solution on large graphs.

Also for ER graphs, it is harder to extraplate to smaller graphs. As we fix the edge probability for different sizes of the graphs, the node degree distribution would shift when graph size changes. This may prevent GNNs from extrapolating well.

## F.3 MORE ABLATION RESULTS

Table 6: Ablation: Real world MVC and MDS relative error.

| Data | MVC | | | | MDS | | | |
|---|---|---|---|---|---|---|---|---|
| | Raw | Onehot | RandFeat | DAD | Raw | Onehot | RandFeat | DAD |
| Collab | $1.50e^{-4}$ | $2.05e^{-4}$ | $3.20e^{-4}$ | $\mathbf{1.28e^{-4}}$ | **0** | **0** | $6.00e^{-3}$ | **0** |
| Twitter | $2.58e^{-3}$ | $2.40e^{-3}$ | $1.23e^{-2}$ | $\mathbf{1.93e^{-3}}$ | $8.67e^{-3}$ | $1.19e^{-2}$ | $3.37e^{-2}$ | $\mathbf{8.42e^{-3}}$ |
| Github | $3.15e^{-5}$ | $2.86e^{-5}$ | $1.39e^{-5}$ | $\mathbf{1.09e^{-5}}$ | $\mathbf{2.55e^{-6}}$ | $1.22e^{-5}$ | $2.25e^{-5}$ | $4.14e^{-5}$ |
| Facebook | **0** | **0** | **0** | **0** | **0** | **0** | **0** | **0** |
| Memetracker | $\mathbf{7.94e^{-6}}$ | $1.59e^{-5}$ | $3.49e^{-4}$ | $1.55e^{-5}$ | $1.94e^{-5}$ | $\mathbf{8.47e^{-6}}$ | $5.21e^{-4}$ | $7.67e^{-5}$ |

Table 7: Ablation: Real world MAXCUT.

| | Raw | Onehot | RandFeat | DAD |
|---|---|---|---|---|
| Optsicom-B | $1.14e^{-2}$ | $3.49e^{-5}$ | $9.92e^{-3}$ | $\mathbf{4.23e^{-3}}$ |
| Optsicom-G | $1.15e^{-4}$ | $8.79e^{-3}$ | $2.93e^{-5}$ | $\mathbf{1.16e^{-5}}$ |

Here we include more ablation results, on both real world datasets (see Table 7 and Table 6) and synthetic datasets (see figure 12). We can see the conclusion is still consistent with our claim in the main paper.

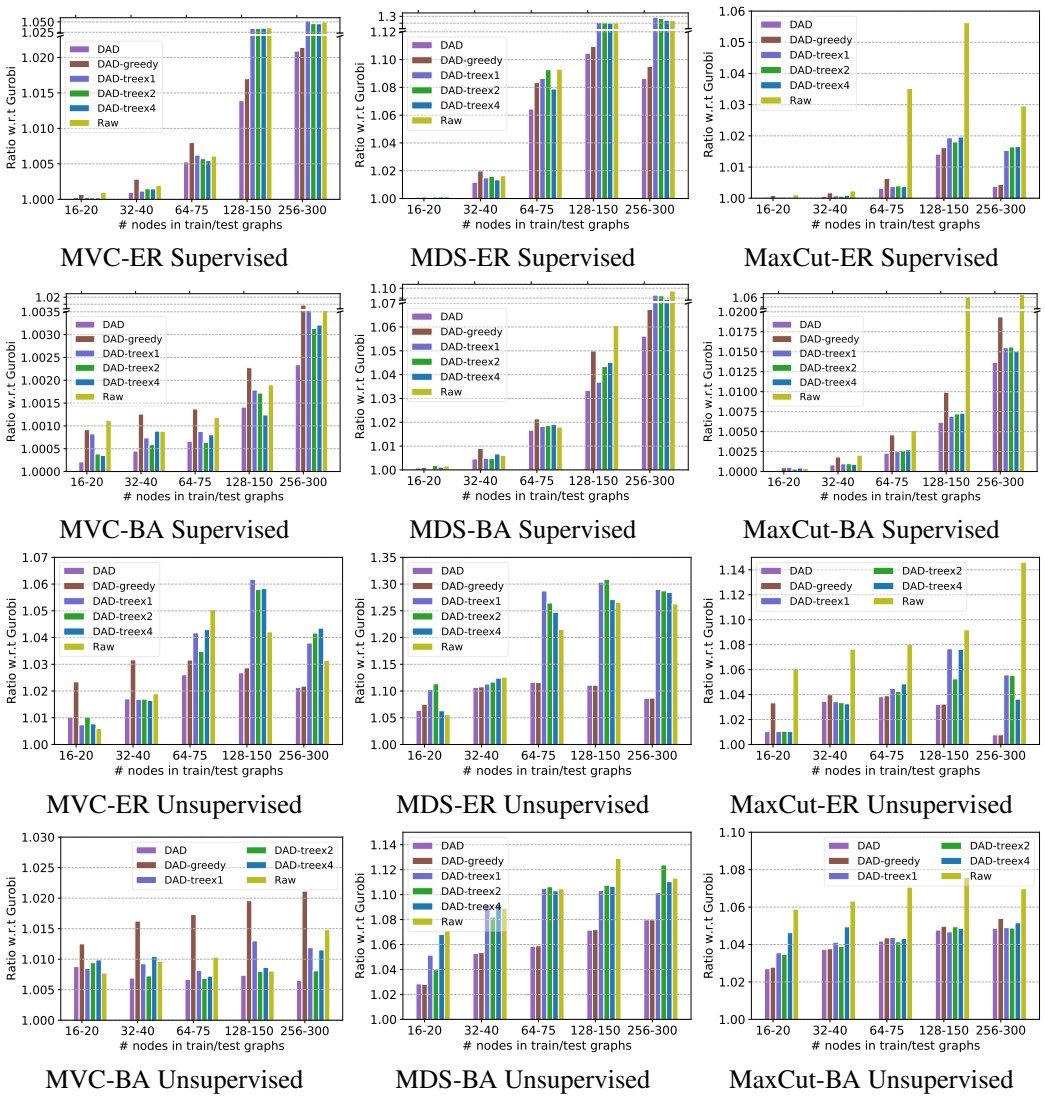

Figure 12: Ablation studies on synthetic graphs. ER stands for Erdos Renyi random graphs, and BA stands for Barabasi Albert graphs.

### F.3.1 ABLATION ON FEATURE DIMENSION

As our algorithmic feature augmented GNNs have more dimensions in the node and edge features, the input layer thus has more parameters than some of the baseline features. For a more rigorous comparison, we have padded the baseline features to the same dimension, and thus all the methods would have same neural network capacity.

Specifically, for the `Raw` baseline, we simply pad constant 1 (or edge weights, when applicable) to the same dimension as DAD. For `RandFeat` baseline, we augment additional random features to match the dimensionality. The two variants are denoted as `Raw-Multi` and `RandFeat-Multi`.

Figure 13 shows the results across three tasks, with either supervised or unsupervised setting, on two types of random graphs. We can see for some cases the `RandFeat-Multi` would improve a bit over `RandFeat`, but overall the `-Multi` variant is comparable as its original counterpart, while being worse than DAD in all cases. This confirms that DAD's gain of performance is not mainly coming from larger networks, but indeed the algorithmic features improve the global information representation quality.

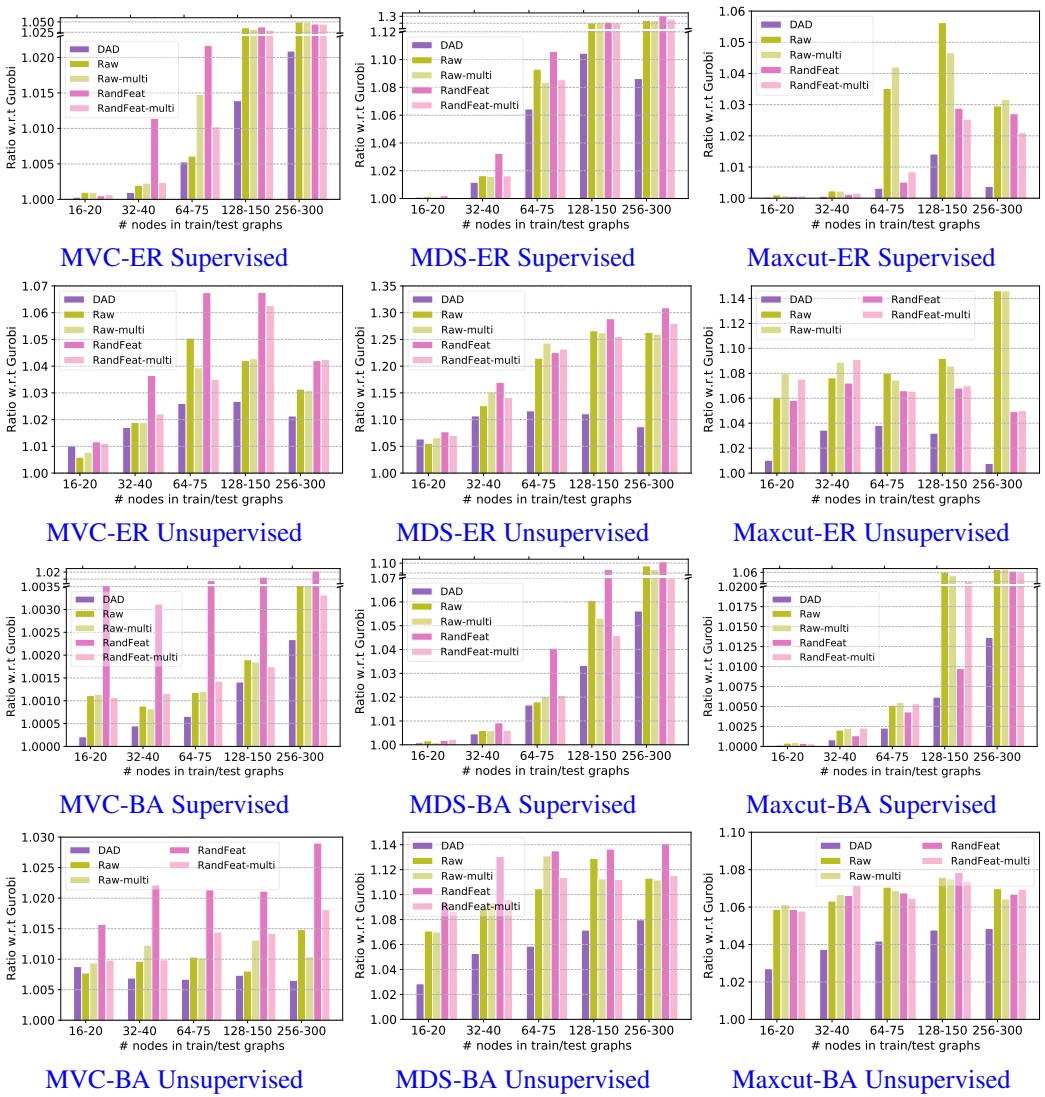

Figure 13: Test approximation ratio by padding baseline features into same dimension.

### F.3.2 MORE TIME-SOLUTION TRADE-OFF PLOTS

We here include the time-solution trade-off plots for all 3 tasks on 2 types of graphs. We vary the average size of graphs in $\{256, 512, 1024\}$. All the evaluation is done using 16 threads, where the baselines are run using embarrassingly parallel setup (*i.e.*, the average runtime is divided by 16), and DAD runs 16 CPU + 1 GPU mixed parallelism. From Figure 14, we can see DAD consistently achieves good trade-off between the time and solution quality.

## G COMPARISON WITH GNNEXPLAINER

In this section, we compare the explanation quality against the GNNExplainer. Without loss of generality, we carry out the explanation experiments on the GNN that is supervised trained for BA graphs with 32-40 nodes, for MVC task.

For GNNExplainer, we follow the implementation from pytorch geometric. We set the entropy regularizer coefficient to be 1, and tune the coefficient for the sparsity regularizer in $\{0.1, 1.0, 10.0\}$. We learn the node and edge masks using Adam optimizer, for at most 3000 gradient update steps.

We visualize the explanation provided by DAD and different configurations of GNNExplainer in Figure 15. In the figure, the node with red boundary color is the target node to be explained. For DAD, we use blue color to indicate the selected nodes or edges, and gray color to indicate the un-selected ones. We fix the explanation budget to be 5 nodes and 10 edges for DAD. For GNNExplainer, the darkness of the color indicates the score of the soft selection.

We can see all the explainers can at least select the target node as the important node when explaining the target node. This at least shows that the explainers are doing reasonable jobs. We can also see from the figure that, DAD can offer some node degree-based explanations for the first and second rows, while GNNExplainer failed to do so, as it either has only one edge with outstanding score, or has a hard time balancing the number of edges with the explanation fitness. Also, from the last row, we can see DAD is able to select some disconnected anchor nodes (see section 5.4) for explanation while GNNExplainer needs either too large or too small chunk of connected subgraphs.

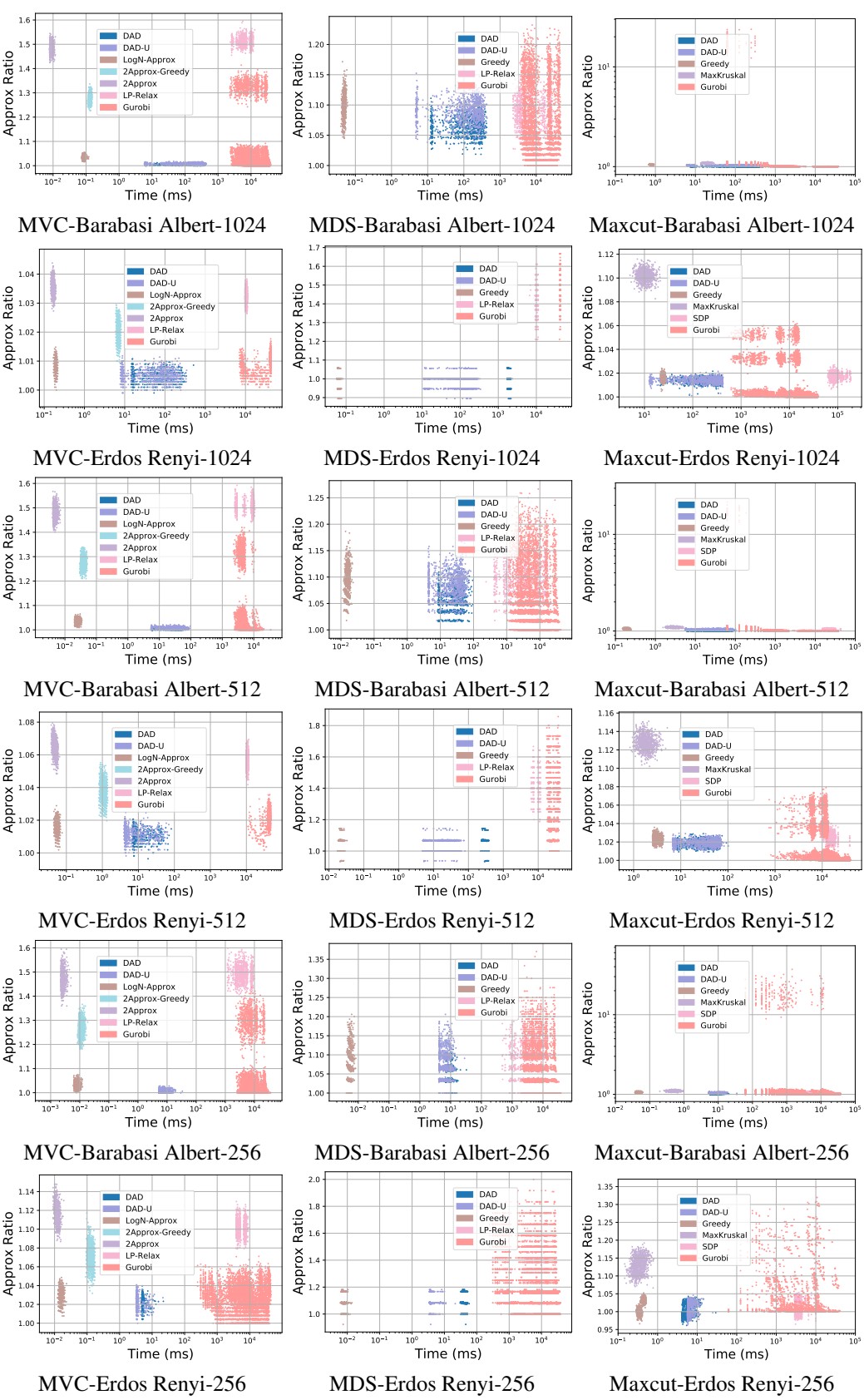

Figure 14: Time and approximation ratio trade-off plot for different methods.

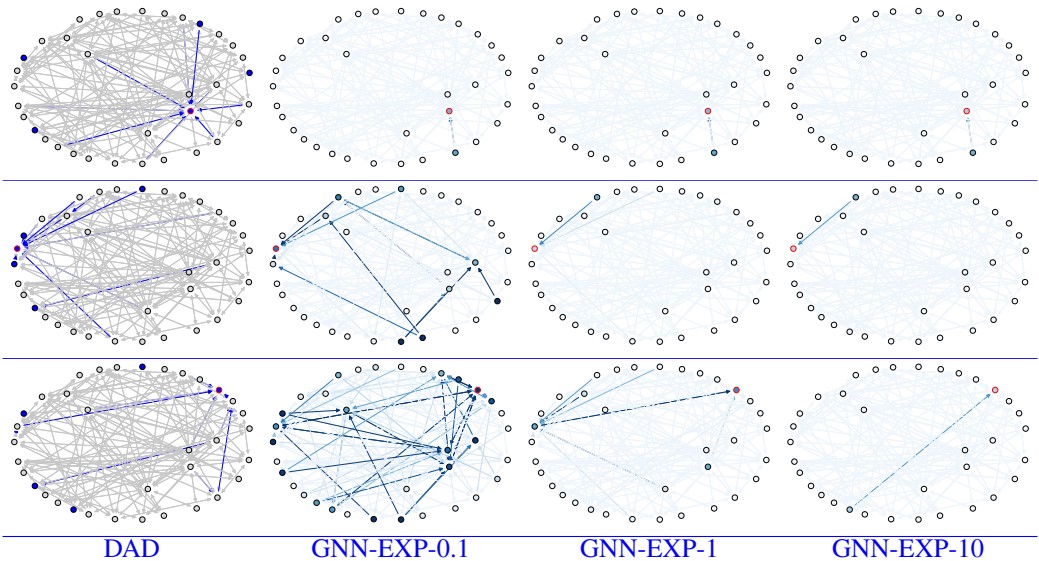

<p align="center">DAD      GNN-EXP-0.1      GNN-EXP-1      GNN-EXP-10</p>

Figure 15: Comparing explanation results with GNNExplainer using different sparsity regularization coefficients.

