# OpenReview forum: "A Framework For Differentiable Discovery Of Graph Algorithms"
_ICLR.cc/2021/Conference — Reject_

### Official Review · AnonReviewer3 · 2020-10-25
**The paper is well presented with detailed motivations and analysis, but has weakness in experiments**

**Rating:** 7
**Confidence:** 3

**Review:**

In this paper, the authors proposed a framework for differentiable graph algorithm discovery (DAD). The framework is developed by improving two the discovery processes, i.e., designing a larger search space, and an effective explainer model. To enlarge the search space, the proposed DAD augments GNNs with cheap global graph features, which consist of solutions on the spanning trees of the original graph, and approximate solutions of greedy algorithms. All the features are concatenated together as input to GNNs. Experiments indicate that the proposed DAD is better than the approximate solutions. The explainer model is developed to explain the learned GNN, by employing the learning to explain (L2X) framework in (Chen et al., 2018). In general, the paper is well-written, and the method is interesting.

Some questions and comments:

1.  Although the authors have provided an ablation study to verify the contributions of tree solution versus greedy solution in Figure 11 in appendix, it is still not very clear about the impact of the global features on the model. The ablation study is short and lacks detailed discussions. In Figure 11, what is "raw" model? The global features are concatenated to the input, which actually increases the dimension of features. It would be necessary to see whether the global information or the extra degrees of freedom improve the performance.

2. How to determine how many spanning tree solutions or greedy solutions is enough for performance improvement? The authors said that "We tuned the hyperparameters of GNN models for each graph category, such as the number of layers and the number of spanning trees, using grid search. " It sounds like the proposed model is hard to generalize to different datasets.

3. The novelty of the explainer model is not very clear. It seems that the authors just applied the learning to explain (L2X) framework in (Chen et al., 2018) to the GNN settings.

4. There many typos, e.g.,
(1) In page 4, "treat the l constraints seperately for efficient computations",    --->  separately
(2) In page 5, above Eq.(11), "is irrelavent to nodes of distrance larger than T"   --> irrelevant, distance
(3) In page 5, "is defined to be the origianl GNN"   --> original
(4) In page 6, "add a global read-out functionto"   --> function


I have read the authors' response, and I would like to keep my current rating.

---

> ### Author Response · Authors · 2020-11-23
> **Reply to Reviewer 3**
>
> Thank you for your constructive comments! We have addressed them accordingly in the paper. Please also see our detailed response below:
>
> ### Q1.1 “ablation...not very clear...In Figure 11, what is “raw” model?”
> ---
>
> The ablation tries to answer which of the following configuration is the best:
> 1. Use both the tree algorithm and the greedy algorithm.
> 2. Use the tree algorithm only (also varying the number of trees).
> 3. Use the greedy algorithm only.
> 4. Use none of the algorithmic features (the “raw” model). We have updated **Sec 5.1** to make it clear.
>
> The above configurations only differ at the graph feature level. They use the same GNN configuration, learning method, and other hyperparameters. Our conclusion is that 1. is the best and is consistently better than 4. This shows that the global feature obtained by 1. is useful.
>
> ### Q1.2 “...whether the global information or the extra degrees of freedom improve the performance…”
> ---
>
> This is a good point. To verify this, we have included **additional experiments in Appendix F.3.1**. To match the number of parameters in the neural network, we expanded the dimension of features for baselines. Specifically, for the raw-configuration, we padded the node and edge features by constant (or edge weights, when applicable); for the random feature baseline, we use the same number of random node and edge features (or edge weights, when applicable).
>
> We found that adding additional constant dimensions didn’t help, while adding additional random features helped the RandFeat baseline in some cases to catch up with the ‘Raw’ baseline. However, the proposed DAD is still consistently better in all 3 tasks, 2 types of random graphs, and both supervised/unsupervised settings. This confirms that DAD’s gain of performance is not mainly coming from larger networks, but indeed the algorithmic features improve the global information representation quality.
>
> ### Q2 “how to determine how many ... tree ... or greedy... ”
> ---
> Usually, greedy features don't have randomness, so we only use one solution per each greedy algorithm. As spanning trees may not be unique and we can use more than one tree into our algorithm, we generally found the more trees we use, the better the algorithm performance is in the end. However, the gain diminishes. So empirically, four spanning trees would be enough for our experiments. This is consistent across different datasets (synthetic, real-world) and different tasks (MVC, MAXCUT, MDS).
>
> ### Q3 “The novelty of the explainer model is not very clear…”
> ---
> The explainer follows the information theory principle from L2X, with several technical distinctions:
> - We want to explain with the discrete subgraph. This is different from discrete feature selection in L2X, and the continuous GNNExplainer formulation.
> - We have an explicit budget constraint for explanation (number of nodes and edges) to make it accessible for humans. L2X uses Gumbel-Softmax, which will have a high bias when stacked multiple times. We novelly developed a differentiable topK operator based on the latest smoothing technique that is efficient and has a low bias.
>
> We mainly focused on the learning of new graph combinatorial optimization algorithms. The explainer can have its own independent interest, but it mainly serves the need for interpreting the algorithm we have learned.
>
> ### Q4 “...typos…”
> ---
> Thanks a lot for pointing out this! We have run a grammar checker and fixed all the issues we found.

---

### Official Review · AnonReviewer1 · 2020-10-26
**Novel approach to explain DNN-based graph algorithms, but experiments can be improved and the explainer model seems flawed.**

**Rating:** 4
**Confidence:** 3

**Review:**

Summary:
This paper proposes a framework to discover graph-algorithms that are learned by neural networks to solve combinatorial optimization problems. To this end, the authors propose (a) augmenting the input of DNN-solver using features extracted by existing combinatorial algorithms and (b) explaining the DNN based on an additional "explainer" model trained based on maximizing the lower bound of mutual information between subgraph of the input and the labels predicted by the DNN solver. I think this paper pursues an important and promising direction to extract algorithms from DNN-based solvers. However, I think (a) additional baselines should be incorporated for evaluating the DNN-solver, (b) the proposed explainer does not generate practically useful outputs for discovering new algorithms, and (c) the proposed explainer seems a bit flawed.

Strength:
- The proposed augmentation scheme gives interesting insights and can be applied to other DNN-based solvers for combinatorial optimization.
- This paper tackles an interesting problem of discovering new algorithms from DNN-based solvers for combinatorial optimization.

Weakness:
- It is not clear why the authors only consider baselines with polynomial-time running time. To show that the proposed DNN-based solver is practically useful, the authors should compare with state-of-the-art solvers (both DNN-based and non-DNN-based) under limited running time. To compare the algorithms based on the tradeoff between complexity and approximation ratio, the authors should provide a theoretical analysis of the approximation ratio of the proposed algorithm.
- It is not clear how to use the proposed algorithm for discovering new graph algorithms. Especially, the algorithms produce results that are not very "explainable." For example, how did the authors use the explainer to "re-discover the node-degree greedy algorithm?" The proposed framework seems to assume that humans can easily analyze the provided explanations (i.e., graphs with colored nodes). However, it seems hard for me to analyze the pattern in node-degree of nodes just by glancing at several explanations provided by the proposed framework. Such a process becomes especially harder if we aim to discover algorithms based on novel concepts (instead of node-degree). Even more, researchers are usually interested in developing algorithms for large-scale graphs (where exact solutions are intractable), yet the explanation becomes notoriously hard to analyze in the eyes of humans for this case. The authors are encouraged to provide an actual process of extracting analysis on the explanations, e.g., did they (1) look at hundreds of explanations provided by the explainer, (2) intuitively group explanations with the common pattern in node-degree, and (3) infer a greedy pattern in node-degree of the selected nodes?
- The proposed explainer seems flawed for explaining the DNN-based solver. Namely, the proposed explainer only accesses the DNN-based solver based on its prediction probability. However, I do not think it makes sense to rely only on the prediction to explain the black-box algorithm. To demonstrate, both the brute force search and integer programming solver will give an identical prediction (i.e., optimal solution) for the combinatorial optimization. Applying the proposed explainer to two algorithms would give an identical explanation, but brute force search and integer programming operate in a very different way.

---

I appreciate the thoughtful rebuttal provided by the authors. My main concerns are on Q2, i.e., the practical usefulness of the algorithm. I do think that the authors provide a convincing argument on "we can only understand what we can understand," hence we should set up a hypothesis and see if it aligns with the explanation. However, I think the usefulness is not well-supported in the current state of the paper. The authors can come up with (a) a stronger example of such a hypothesis and (b) a better measurement of how the hypothesis aligns with the explanation to strengthen the paper.

Regarding Q3, I still think that it is not correct to provide the same explanation for different algorithms when they produce the same output. Hence, the proposed algorithm should be modified to consider this aspect.

---

> ### Author Response · Authors · 2020-11-23
> **Reply to Reviewer 1, part I (Q1.1, Q1.2)**
>
> Thank you for your insightful comments! We have added additional experiments and refined the paper as requested. Please see our detailed response below:
>
> ### Q1.1: “compare with state-of-the-art solvers (both DNN-based and non-DNN-based)”
> ---
> We have already compared with three recent DNN-based papers [1][2][3] in Fig 3. All of them were published after 2019, and [3] has just got accepted in the upcoming NeurIPS 2020.
> We have also compared against the best commercial solver -- Gurobi. All the approximation ratios in our experiment are calculated against the best solution obtained by Gurobi with one hour time budget.
>
> Note that for these NP-Complete (NPC) problems, the exhaustive search can already get the optimal solution. So what matters the most is the time - solution quality trade-off. As our main goal is to improve GNN’s expressiveness for NPC problems, it is more comparable to algorithms in the polynomial family. Nevertheless, we have included additional comparison results in the revised paper. Specifically:
>
> We show the **additional results in Fig 4** on BA graphs with minimum 1024 nodes. For each single graph, we record the running time and the corresponding approximation ratio, and obtain a dot in the scatter plot. As Gurobi solver is an iterative process, we plot the data points every time it improves the solution on the graph, and truncate. We run baselines with 16 CPU cores in parallel over 1000 graphs, as we use 1 x GPU for DAD. As illustrated in the Fig 4, DAD can achieve similar or better solution quality than what can be achieved by Gurobi, with 2 magnitudes less time. Although the time comparison is not exactly fair due to implementation factors, it is still safe to conclude DAD achieves good time-solution quality trade-off.
>
> ### Q1.2: “theoretical analysis of the approximation ratio”
> ---
> We have included the theoretical analysis in the paper as suggested. Our analysis largely follows [1], which studies the expressiveness of distributed local algorithms in the scenario of approximating the NPC problem.
>
> -  We show that our Tree-DP feature is a generalization of the idea in [1]. For example, in MAX-CUT, DAD will first obtain a solution on the spanning tree, where the Tree-DP solution itself is the 2-coloring of the tree, and thus the weak 2-coloring of the original graph. By invoking [1], we can achieve an approximation ratio of 2.
>
> - Our framework also encodes the greedy algorithm solution as features. For MVC, the greedy algorithm we used can have ratio 2, or O(log N) that grows with N; For MDS we adopt the O(log N)  ratio. As GNN on top of the node feature is a distributed local algorithm, it admits the identity function. Thus, it is straightforward to show that it can achieve ratio=2 for MVC, and O(log N) for MDS.
>
> Note that the above analysis considers the expressiveness of DAD only. The PAC learnability analysis is beyond the scope of this paper and will be interesting for future work.
>
> [1] Approximation Ratios of Graph Neural Networks for Combinatorial Problems, NeurIPS 19.
> [2] Random Features Strengthen Graph Neural Networks, 2020
> [3] Erdos goes neural: an unsupervised learning framework for combinatorial optimization on graphs, NeurIPS 2020

---

> ### Author Response · Authors · 2020-11-23
> **Reply to Reviewer 1, part II (Q2, Q3)**
>
> Thank you for your insightful comments! This is part II of our reply. Please kindly refer to the thread below for the first part of our reply to your first set of questions.
>
> ### Q2: “not clear how to use the proposed algorithm… for large-scale graphs ... actual process of extracting analysis … ”
> ---
>
> **Novel concept and going beyond:**
> This is an interesting dilemma, where we can only understand what we can understand. So the reasonable way to approach this is to form the hypothesis we can understand and verify if it aligns -- a typical scientific way of understanding physics/chemistry. Generally, it is up to the ‘vocabulary’ of the explanation space in our DAD framework. One way to make the explanation even more interpretable is to further restrict the space of explanation to even simpler patterns. At the moment, we allow the explainer to select completely different patterns for each node, which is equivalent to a very flexible explanation space. However, we can restrict the explanation space to a small vocabulary of subgraph patterns, and restrict the explanation selection to either select a subgraph for all nodes or not. Such restricted space of selection typically can improve the interpretation for human, but it will potentially also lead to reduced performance for the algorithms which use these selected subgraphs only. There is an interesting trade-off between the flexibility of the explanation space and the effectiveness of the algorithms leveraging these selected patterns. We have not seen a dedicated treatment on this subject in the GNN explanation literature, and we will leave an in-depth investigation of this subject as a future work.
>
> **Large graphs:**
> Because the same graph neural network can be applied to graphs of arbitrary sizes, usually, it is reasonable to observe the explanation for small graphs and generalize it to larger graphs, since the GNN (so as the understanding) trained on small graphs typically extrapolates to large graphs, as we have shown in Fig 10, 11.
> Also, as we can explicitly control the actual number of nodes/edges for explanation, we can balance between the difficulty of explanation and the faithfulness to the predictive quality.
>
> **Actual process:**
> The plot we obtained in the paper generally requires human inspection as the reviewer mentioned. We first observe patterns in the explanations, then form our hypothesis, and next verify with some similar ideas that we have. For example, the anchor nodes idea we observed are from recent position aware GNN papers; the degree based hypothesis comes from greedy algorithms, etc. Obtaining these patterns for small graphs is tractable, especially given that our explainer is restricted to provide sparse subgraph (5 nodes, 10 edges) explanations.
>
> ### Q3: “...do not think it makes sense to rely only on the prediction to explain the black-box algorithm..”
> ---
>
> We thank the reviewer for raising this interesting argument. At the moment, our explainer model indeed focuses on only explaining the final decision made by the learned GNN algorithm, which is just one aspect of explaining an algorithm. As far as we understand the question, the reviewer is asking the possibility of explaining each step of a learned GNN algorithm.
>
> We think our explainer model can be adapted to explain each step of the algorithm. This is possible because a learned GNN algorithm is a highly structured and biased neural algorithm. More specifically, a GNN algorithm is an iterative distributed local algorithm where each step of the algorithm is determined by a local message passing operator parameterized by a neural network with pooling operation. Essentially, a GNN algorithm leverages this learned local message passing operator to gradually identify local graph patterns to arrive at the final decision for a center node. Such structure of the GNN algorithms have been discussed extensively in recent literature. For example,
> - Sato et al. Approximation Ratios of Graph Neural Networks for Combinatorial Problems. NeurIPS 2019.
> - Xu et al. How powerful are graph neural networks. ICLR 2019, which connects GNNs to graph isomorphism test algorithms such as Weisfeiler-Lehman algorithm.
>
> Thus, to achieve explanation of the individual steps of a learned GNN based distributed algorithm, we can explain the input-output relation of the local message passing operator. Since the message passing operator typically only involves a node and its immediate neighbors, we can adapt our explanation techniques to explain the operator using the one-hop neighborhood of a node. A detailed experimental study of this aspect of explaining the algorithm step will be left for future study.

---

### Official Review · AnonReviewer4 · 2020-10-26
**neat idea, trustworthy results**

**Rating:** 6
**Confidence:** 3

**Review:**

the paper has provided an explainable GNN framework using differentiable graph discovery algorithm. To me more specific, it utilize the solution over spanning trees greedy approximation and the explainer GNN is able to provide the influential node/edges. The experimental results has demonstrated the effectiveness of the proposed framework.

 The idea of the model is quite neat and artful designed and it is also intuitive formulate as ILP problem. However, I do have some concerns as follows:

1) In terms of the explainable GNN, I doubt the presenting systematic explanation of the graph model is explainable. It lacks of the details of perspective of explainability/interpretability definition over in either quantitative manner or qualitative way(e.g. user study/case study in downstream task). It could be still regarded as an open problem in AI transparency and I'm not criticizing this paper has not provided the corresponding merits. However, explaining the graph models leveraging as node/edge selections probably not uniform solution but a neat idea.

2) The paper also mentions the work provided by GNNExplainer, it would be more confident to believe your proposed method is optimal if considering the GNNExplainer as baseline. More importantly, it would be essential to reason why your proposed framework is better in terms of explainability.

---

> ### Author Response · Authors · 2020-11-23
> **Reply to Reviewer 4**
>
> Thank you for your constructive comments! We have added additional experiments as suggested and modified the paper accordingly to address your questions. Specifically:
>
> ### Q1: "whether explainable...leveraging as node/edge selections probably not uniform solution.. "
> ---
> Thanks for raising this interesting discussion. We agree that it is hard to explain in a way that is fully end-to-end interpretable for humans. Our framework proposes an algorithm space design + interpretable distillation schema, where one can potentially use more interpretable models with human inductive bias in this framework.
>
> The node/edge selection mainly obtains a sparse structure that would be more accessible for humans. Based on this principle, one can extend it to other interpretation spaces. In our **Sec 5.4** we have included the study of algorithmic feature importance.  There can be more extension. For example, one could define a collection of graphlets or interpretable graph patterns and then select the most influential graphlets or patterns, using similar optimization techniques introduced in our paper. However, we could see that there is a  tradeoff between the interpretability and the design of the interpretation space, where node/edge selection requires the least effort in the design.
>
> Please also kindly refer to our answer to Q2 from Reviewer 1 for more explanations. Thank you!
>
> ### Q2: "...GNNExplainer as baseline … why your proposed framework is better..."
> ---
> Firstly, there are several requirements in our framework that GNNExplainer does not satisfy, which motivates us to seek for new solutions, specifically:
>
> - a) We want to discover the explainable polynomial algorithm, which is able to perform inference as well as explanation on new instances. However, GNNExplainer optimizes an explanation mask for **each** single node of **each** graph, which is no longer a polynomial algorithm and is time-consuming.
> Furthermore, optimizing for individual graphs may lack the global consistent understanding of how the model behaves for the distribution of graphs.
> -  b) We want to obtain discrete/hard subgraphs, while GNNExplainer offers continuous/soft subgraph selection. Although post-processing is possible, it is tricky. Our end2end discrete interpreter would be more preferable.
> -  c) It only makes sense for GNNExplainer to extract subgraphs that are connected to the target node, as non-connected components won’t have influence due to their GNN parameterization. Instead, we introduce a variational model that allows using separated subgraphs for explanation, which enriches the explanation space.
>
> We have included **additional experiments in Appendix G** to verify that the technique we developed is better in terms of a)b)c) listed above. Our conclusion is:
>
> - For a), our method only needs one feed-forward pass to obtain the sparse graph structure, while for GNNExplainer, we optimize the mask for 3000 steps, which roughly means **1:3000** regarding the runtime comparison.
> - For b), it is a bit subjective, but we show that the fuzzy continuous map is somewhat hard to interpret, and also tuning the sparsity penalty term for GNNExplainer is tricky.
> - For c), we are able to find the anchor nodes with the proposed technique, while GNNExplainer has to include unnecessary nodes in between to make it connected.
>
> Actually, we have also noticed a concurrent paper named “Parameterized Explainer for Graph Neural Network” which was posted on arxiv on Nov 9, 2020 and accepted in NeurIPS 2020. This paper shares similar motivations as ours, despite that our optimization technique is different. We have also included the discussion in Appendix A.

---

### Decision · Program_Chairs · 2021-01-07
**Final Decision**

**Decision:**

Reject

**Comment:**

This paper proposes a method for automatically discovering graph algorithms using GNNs. In general, the reviewers find the paper well-written, and the problem and the approach interesting.  However, there is a concern on the practical usefulness of proposed method as shown in the following comments:  “My main concerns are on Q2, i.e., the practical usefulness of the algorithm”[R1]; “It sounds like the proposed model is hard to generalize to different datasets” [R3]; “The proposed explainer does not generate practically useful outputs for discovering new algorithms”[R4].